# A Hybrid Price Auction-Based Secure Routing Protocol Using Advanced Speed and Cosine Similarity-Based Clustering against Sinkhole Attack in VANETs [note 1]

**DOI:** 10.3390/s22155811

**Published:** 2022-08-03

**Authors:** Yushintia Pramitarini, Ridho Hendra Yoga Perdana, Thong-Nhat Tran, Kyusung Shim, Beongku An

**Affiliations:** 1Departement of Software and Communications Engineering in Graduate School, Hongik University, Sejong City 30016, Korea; yushintia@gmail.com (Y.P.); mail.rhyp@gmail.com (R.H.Y.P.); 2Departement of Electronics and Computer Engineering in Graduate School, Hongik University, Sejong City 30016, Korea; trantnhat@gmail.com; 3Departement of Software and Communications Engineering, Hongik University, Sejong City 30016, Korea; shimkyusung@outlook.kr

**Keywords:** secure routing, clustering, auction, security, sinkhole attack, vehicular ad-hoc networks

## Abstract

In ad-hoc vehicle networks (VANETs), the random mobility causes the rapid network topology change, which leads to the challenge of the reliable data transmission. In this paper, we propose a hybrid-price auction-based secure routing (HPA-SR) protocol using advanced speed and cosine similarity-based (ASCS) clustering to establish a secure route to avoid sinkhole attacks and improve connectivity between nodes. The main features and contributions of the proposed HPA-SR protocol are as follows. First, the HPA-SR protocol is employed by the first- and second-price auctions to avoid sinkhole attacks. More specifically, using the Markov decision process (MDP), each node can select a kind of auction method to establish the secure route by avoiding the sinkhole attack. Second, the advanced speed cosine similarity clustering protocol that is considered as underlying structure is presented to improve the connectivity between nodes. The ASCS is constructed based on the cosine similarity and distance between nodes using the speed and direction of the nodes. The results of the performance show that the proposed HPA-SR protocol can establish the secure route avoiding the sinkhole attack while the proposed ASCS clustering can support the strong connectivity. Besides, the HPA-SR with ASCS protocol can show better performance than the benchmark protocol in terms of the routing delay, packet loss ratio, number of packet loss, and control overhead.

## 1. Introduction

The development of hardware and wireless communication techniques can support communication between vehicles or vehicles and roadside units (RSU) [1,2]. However, since vehicles can dynamically and independently move, the direct transmission is very challenging. Thus, routing protocol is considered as one of the possible solutions to establish the route in vehicular ad-hoc networks (VANETs) [3,4,5]. In addition, the clustering protocols can enhance the stability of networks by making the clustering and electing the cluster head that can communicate with different cluster heads [6]. Thus, if the cluster exists in the network, the route connectivity is more stable than the network without clustering [7]. Since the centralized (managed) node does not exist in the VANETs, e.g., access point and base station, it is very vulnerable to the networking attacks. For example, denial of service (DOS) attack, blackhole attack, wormhole attack, sinkhole attack, etc. [8].

In VANETs, vehicles driving along roads can be formed into groups to facilitate communication. Clustering stabilizes the connectivity between nodes by hierarchizing the network architecture, where each cluster can be divided into three kinds of nodes, called cluster head (CH), cluster member (CM), and gateway (GW) [9,10]. When CM needs to communicate with other nodes, CM sends the message to its CH. This CH can communicate with other CHs. If the intermediate CH is out of transmission coverage, the GW helps to relay between CH and CH. When the data packet arrives at the last CH, the CH sends message to the desired CM.

Routing protocols play an essential role in ensuring reliable data communication [11]. The routing protocol can select the suitable next nodes from a source node to a destination node among intermediate nodes. Routing in a VANET is complicated due to its dynamic nature including significant mobility and various network topologies. Therefore, routing protocols designed for the VANET environment must meet the standards of scalability, efficiency, and comparability. Recently, the security has become one of the critical issues in the network [12]. Considering the accessibility of the network and also cyber security has drawn much attention for cyber-physical systems (CPS) [13]. Moreover, blockchain [14,15] is one of the solutions when it comes to supporting credible distributed communication, and basically having the main characteristics of decentralization, distributed consistency, etc. In VANET, there are three types of attacks. The first type is an attack on the communication infrastructure, such as routing protocols, the second type is an attack on VANET’s functions, such as location monitoring, and the third type is an attack on security requirements, such as authentication protocols. The sinkhole attack is one of the network threats in which the sinkhole node announces itself as having the best path to the sink or destination node and prevents data transmission between a source and a destination node by broadcasting fake routing information [16,17]. During the routing process, the sinkhole node can provide fake information. Therefore, the data packet sent by the source node does not arrive at the destination node.

## 2. Related Work and Motivations

### 2.1. Related Works

Due to high mobility in VANETs, it brings the disconnection between nodes in urban scenarios. Therefore, we require a clustering protocol to achieve high stability as well as reducing control overhead. The author in [18] proposed a multi-hop broadcast protocol to efficiently disseminate emergency warning messages in a VANET with highway scenario. The authors in [19] proposed the lowest ID (LID) and highest degree (HD) algorithm. In the LID algorithm, the node with a lower identifier becomes a cluster head. In the HD algorithm, the node with the highest number of neighbors is selected as CH. The authors in [20] addressed the passive multi-hop clustering algorithm (PMC) to ensure the coverage and stability of the clusters. In [21], the authors studied the vehicle selection based on sigmoid function in which the vehicles with large object functions are selected as cluster heads. However, the proposed algorithm does not consider stability and connectivity between vehicles. Hence, to improve the network stability and connectivity, the network parameters that affect on the clustering and cluster head election is considered as the cost for clustering.

Various routing protocols have been proposed secure routing to tackle the node’s strong mobility. The author in [22] addressed an intelligent opportunistic routing algorithm for wireless sensor networks and applies it for e-healthcare. The author in [23] presented an intelligent trust sensing scheme with metaheuristic-based secure routing protocol for Internet of Things in [23], to identify the next hope by using a fitness function. However, it cannot avoid the attacker directly in order to select the next hope. The author in [24] proposed a blockchain-based secure routing protocol for opportunistic networks. In [24], only registered nodes can forward hello packets. If there is a new node, the node must be registered first, then it can forward the hello packet. Otherwise, the node cannot join the network. The author in [25] studied a trust-based secure intelligent opportunistic routing protocol to avoid the gray and black-hole attack in the wireless sensor networks. However, these works [22,23,24,25] proposed secure routing that did not focus on sinkhole attack in VANET. The authors in [26] addressed the method of speed adaptive beacon broadcast (SABB) to propagate information in urban and highway environments. In [27], the authors exploited the vulnerability attacks of distributed vehicular broadcast (DV-CAST) protocol and pointed out safety specifications that are necessary for DV-CAST security against specific attacks. Several studies had implemented sinkhole attack detection and prevention on networks. The authors in [28] proposed an individual trust managing (ITM) technique to prevent against sinkhole attack in MANETs. The authors in [29] presented an approach to prevent the sinkhole attack in mobile ad hoc networks (MANETs). In [29], hop count-based detection techniques is used to detect sinkhole attack. The proposed solution to detect the sinkhole node utilizes the non-propagating route request techniques. The authors in [30] proposed memory effective node collusion method to prevent and detect sinkhole and wormhole attacks using a modified AODV protocol. Su et al., in [31], studied the multi-path multi-hop routing in networks with selfish nodes. The authors in [32] proposed an algorithm based on auction mechanism for vehicle routing problem to ensure that the auction could be used in operational decision making. However, these secure routing protocol in [26,27,28,29,30] established the secure route against sinkhole attack without auction theory, which leads to the additional process to detect the sinkhole node. Different from [26,27,28,29,30], the work [31,32] proposed the auction theory-based secure routing protocols that did not require the sinkhole node detection process for the secure route establishment. Consequently, we can conclude that auction-based secure routing protocol can avoid the sinkhole attack without the detection process, which can reduce the control overhead and delay as well as sinkhole node avoiding. Therefore, the auction theory-based secure routing protocol is one of the possible solutions for VANETs.

### 2.2. Motivation and Contributions

Since the sinkhole attack causes various issues in VANETs, we propose the secure routing protocol to avoid the sinkhole attack. Different from the related works [19,20,21,26,27,28,29,30,31,32], in this paper, we propose a hybrid-price auction-based secure routing protocol (HPA-SR) to avoid the sinkhole attack without detection. The proposed secure routing protocol employs the Markov decision process (MDP) to select the next nodes as well as sinkhole node avoiding, where the MDP can switch from first price auction to second price auction based on the number of routes to avoid the sinkhole attack adaptively. In addition, we propose the mobility-based clustering protocol, called advanced speed and cosine similarity-based clustering (ASCS), to enhance the route stability and reduce the control overhead, which is modified by our previous work [33]. The proposed ASCS clustering protocol can enhance the route connectivity. The main contributions of this paper can be summarized as follows:We propose a novel hybrid price auction-based secure routing (HPA-SR) protocol to avoid sinkhole attacks. More specifically, the proposed HPA-SR protocol contains the first- and second-price auction. Each node employs the Markov decision process to conditionally select which kind of auction method used to establish the secure route against the sinkhole attack without detection.We further propose an advanced clustering protocol, called advanced speed and cosine similarity-based clustering (ASCS) protocol as underlying structure, to improve the route connectivity and reduce the control overhead in VANETs. The proposed clustering protocol consider node speed and direction as cosine similarity and cosine distance to form the clusters. In addition, the ASCS clustering protocol elects gateway nodes to support the communication between CHs when the next CHs is out of the transmission coverages.The performance evaluations show that the proposed routing protocol can establish more robust route against the sinkhole attack compared to that of AODV. Besides, the proposed ASCS clustering supports more strong connectivity since the clustering transforms the network topology hierarchically.

The rest of the paper is arranged as follows. Section 3 introduces the background theorem that consists of the background of auction theory, first-price auction, and second-price auction. Section 4 introduces the proposed routing protocol that consists of the basic concept of the proposed routing protocol, the proposed clustering protocol (ASCS) and the proposed hybrid-price auction-based secure routing protocol (HPA-SR). Section 5 presents the performance evaluation that consists of simulation environtments and parameters, performance metrics, and numerical results. Section 6 concludes the paper.

## 3. The Background Theorem: Auction Theory

### 3.1. The Background of Auction Theory

In this section, we present the process of establishing a secure route between a source and a destination node based on the auction theoretic algorithm. The auction theoretic algorithm belongs to a class of games in which a principal would like to condition the node’s actions on some information that the other player privately knows. We design the hybrid-price auction-based secure routing (HPA-SR) protocol which consists of first- and second- price auction as explained in Section 4.4. Based on our auction model, the proposed HPA-SR protocol can avoid the sinkhole attack without detection and can also reduce the control overhead and delay while establishing a secure route.

In theory, the auction algorithm incorporates buying and selling items into the bid process [34]. In addition, the auction process is often used to sell objects that do not have a fixed or unspecified price. To simplify, we limit a single seller and only sell an item. Thus, the auction procedure can involve

The seller offers only one item for sale,The *i*-th buyer of *N* buyers will have an object valuation ($v_{i}$) with $v_{i}$ ≤ 0.

#### 3.1.1. First-Price Auction

The value of the player’s bid affects whether or not the player wins and how much the player pays in the first-price auction. Thus, the most of the reasons for creating the previous section must be redone, and the conclusions have changed. We can suppose the auction is a game where players are bidders, and each bidder’s strategy is the amount bid as a function of its true value. Suppose the winner of the game is player *i*, whose bid is $b_{i}$. Then, the payoff of player *i* is $v_{i}-b_{i}$ because the player *i* value for the sold object is $v_{i}$. For the other players the payoff is 0. It should be noted that the winner’s payoff can be negative. This occurs when a player wins the object by overbidding or submitting a bid that is higher than her valuation of the object being sold. For two players participating in the auction, such as *i* and *j*, the payoff function of player $p_{i}$ is [35]:(1)$$\begin{array}{c} p_{i}=\left\{\begin{array}{cc} v_{i}-b_{i} & b_{i}>b_{j}\\ 0 & b_{i}\leq b_{j}, \end{array}\right. \end{array}$$
where *i* and *j* presents the two players in the auction, $b_{i}$ is the bid of player *i*, and $v_{i}$ is the value of the auction to player *i*. The theorem in [36] provides a thorough description of its Nash Equilibrium.

#### 3.1.2. Second-Price Auctions

The winner of the second-price auction is the player who submitted the highest bid, but the player pays the seller the amount equal to the second highest bid [37]. If there are no ties in this auction, the winner pays a lower price to the seller than in the first-price auction, and the payoffs are now defined as follows:(2)$$\begin{array}{c} p_{i}=v_{i}-\bar{b}, \end{array}$$
where $v_{i}$ always returns a non-negative payoff but can now produce a completely positive payoff and the highest bid $\bar{b}$ = $\max b_{j}$, j≠i. Note that if $v_{i}<b_{i}$ then there is still a winning curse going on here and some other bids are in the open interval $\left(v_{i},b_{i}\right)$. There are Nash equilibrium of second-price auction:$\left(b_{1},\cdots ,b_{n}\right)=\left(v_{1},\cdots ,v_{n}\right)$, where each player’s bid is equal to the other player’s valuation.$\left(b_{1},\cdots ,b_{n}\right)=\left(v_{1},0,\cdots ,0\right)$, where player 1 gets the object and the other player’s payoff is zero.

In these two equilibrium, we just described a player getting the object. However, there is a equilibria where the player does not get the object, such as $\left(b_{1},\cdots ,b_{n}\right)=\left(v_{2},v_{1},\cdots ,0\right)$, where another player gets an object with a price of $v_{2}$ and each player receives a zero payoff. We denote the second-price auction equilibrium is $\left(v_{1},\cdots ,v_{n}\right),\left(v_{1},0,\cdots ,0\right),\left(v_{2},v_{1},\cdots ,v_{n}\right)$. However, the property suggests that this equilibrium is less plausible as an auction outcome than the first equilibrium, in which each player bids on the valuation of the other player. The last equilibrium’s weakness is reflected in the fact that player 2’s bid $v_{1}$ is weakly dominated by the bid $v_{2}$, as described in [38]. Besides that, it is very difficult to uncover the truth. However, in other cases, when a player bids less than another player, that player will never win. We need to show that when player *i* bids $b_{i}=v_{i}$, no deviation from this bid improves the other player’s payoff, regardless of the strategy used by each player. There are two cases considered—the first case is when a deviation occurs in which *i* raises another player’s bid, and the second case is when *i* lowers another player’s bid.

The equilibrium of the second-price auction can be expressed as
(3)$$\begin{array}{c} \left(b_{1},\cdots ,b_{n}\right)=\left(v_{1},\cdots ,v_{n}\right) \end{array}$$
where the bid of each player is equal to the valuation the player makes of the object. All other strategies of the player are weakly dominated by the player’s action. Truthfulness is a dominant strategy that makes the second-price auction conceptually very clean. Regardless of what the other bidders do, the most honest bidder is the best choice. Then, the second-price auction will be used when the highest bidder is unfair, which makes sense when the highest bidder is overbidding or colluding. Thus, the second price auction can be expressed as
(4)$$\begin{array}{c} \max_{p_{i}}\;v_{i}-\bar{b}\neq 0, \end{array}$$
where $\max_{p_{i}}$ is maximum payoff of the player *i*, then each player *i* has a value $v_{i}$, and the highest bid is $\bar{b}$. In other words, we determine that the maximum payoff of the player must not equal zero.

## 4. The Proposed Routing Protocol: HPA-SR

### 4.1. Basic Concept of the Proposed Routing Protocol

In this subsection, we present the basic concept of the proposed routing and clustering protocol. The proposed clustering (ASCS) protocol considers the speed and direction of the node to elect the cluster head while cosine similarity and cosine distance for forming the clusters (to decide the cluster members) are underlying structures to support stable connectivity between nodes.

The proposed routing protocol (HPA-SR) utilizes hybrid auction to establish the secure route from a source node to a destination node. As we can see in Figure 1, when the node receives various route information from the neighbor nodes, this node utilizes the second price auction to avoid the sinkhole attack. Therefore, the proposed HPA-SR protocol can establish the secure route such as S-${\mathrm{CH}}_{1}$-${\mathrm{CH}}_{3}$-⋯-${\mathrm{CH}}_{k}$-D, which can avoid sinkhole node.

The route establishment process consists of two steps which can be summarized as follows:**Step 1 (Clustering):** In the first step, we perform a clustering process in which all nodes in the network are divided into clusters by using cosine similarity method. We use the position, direction, and speed as parameters to make a cluster form that works as underlying structure. This work is a development of a paper that has been done by the author in [33]. The ASCS clustering protocol considers the gateway node to improve the route connectivity and reduce the control overhead.**Step 2 (Routing):** After the clustering step, using the hybrid auction method, a source node broadcasts the RREQ packet to find a destination node. When the intermediate nodes receive the RREQ packet, they update their routing table and re-broadcast the RREQ packet. When the RREQ packet arrives at the destination node, the destination unicasts the RREP packet. In addition, the sink hole node also unicasts the RREP packet. If the intermediate nodes receive the RREP packet from the different way, the node utilizes the second price auction to avoid the sinkhole attack. Otherwise, the intermediate nodes employ the first-price auction.

### 4.2. The Proposed Clustering Protocol (ASCS): The Underlying Structure

#### 4.2.1. The Basic Concepts of the ASCS

As shown in Figure 2, the proposed clustering protocol [33], called advanced speed and cosine similarity-based clustering (ASCS) protocol as underlying structure, can improve the route connectivity and reduce the control overhead in VANETs. The proposed clustering protocol consider node speed and direction as cosine similarity and cosine distance to form the clusters. In addition, the ASCS clustering protocol elects gateway nodes to support the communication between CHs when the next CHs is out of the transmission coverages. Figure 3 demonstrates the flowchart of the proposed ASCS clustering protocol. The proposed ASCS clustering protocol consists of two sub-subsections, which can be summarized in the following subsection.

#### 4.2.2. The Proposed Clustering Protocol: ASCS

In this sub-subsection, we explain in detail the proposed clustering protocol, named ASCS. We consider two criteria to form the cluster, namely cosine similarity and cosine distance. The procedure for electing the cluster head and forming a cluster (to decide the cluster members) is as follows:**Step 0: Initialization**When the clustering starts, each node turns on and operates independently.**Step 1: Dissemination of Node Information**A node $n_{i}$ estimates its information, such as speed, direction, and location, periodically. To advertise its node information with neighbor nodes, node $n_{i}$ generates **(INFO)** packet and broadcasts the **(INFO)** packet to its neighbor nodes periodically, respectively. **INFO** packet contains the following fields:
$$\begin{array}{c} \langle \mathrm{Type},S_{\mathrm{ID}},D_{\mathrm{ID}},S,\mathrm{Dir}\rangle \end{array}$$
where Type represents packet type, $S_{\mathrm{ID}}$ represents source node ID, $D_{\mathrm{ID}}$ represents destination node ID, S represents *i*-th node’s speed, and Dir represents its node direction ($\theta_{i}$), respectively.**Step 2: Decission of Node Direction**When $n_{i}$ receives **INFO** packet from the neighbor nodes as shown in Figure 2, the $n_{i}$ checks whether the direction is less than the threshold of the neighbor nodes and if it will be cluster head (CH) or not, which is mathematically expressed as
(5)$$\begin{array}{c} {\mathrm{NB}}_{i}^{*}=\left|\theta_{NB_{i}}\right|<\theta_{th_{i}}, \end{array}$$
where $\theta_{NB_{i}}$ is direction nodes and $\theta_{th_{i}}$ is threshold direction of *i*-th node. The selected nodes means that they can move in the same direction.−If ${\mathrm{NB}}_{i}$ = ${\mathrm{NB}}_{i}^{*}$, go to **step 3**.−Otherwise, the packet will be dropped.**Step 3: Election of Candidate Cluster Heads**The candidate cluster heads (CHs) are selected by the slowest node among the two or more neighbor nodes, which is mathematically expressed as
(6)$$\begin{array}{c} i^{⋆}=\arg\min_{i\in {\mathrm{NB}}_{i}^{*}\cup \left\{i\right\}}\left\{s_{i}\right\}. \end{array}$$Since the cluster head is the smallest node speed in the similar direction, this node can provide strong connectivity between the cluster head and the cluster member nodes.−If *i* = $i^{⋆}$, the node $n_{i}$ becomes cluster head, go to **step 4**.−Otherwise, go to **step 6**.**Step 4: Dissemination of Cluster Head Information**If $n_{i}$ becomes the cluster head, to announce to its neighbor nodes, $n_{i}$ generates and broadcasts the cluster head information **(CHI)** packet to its neighbor nodes. The **CHI** packet contains the following fields:
$$\begin{array}{c} \langle \mathrm{Type},S_{\mathrm{ID}},D_{\mathrm{ID}},S,\mathrm{Loc},\mathrm{Dir}\rangle \end{array}$$
where Type represents packet type, $S_{\mathrm{ID}}$ represents source node ID, $D_{\mathrm{ID}}$ represents destination node ID, S represents its node speed, Loc represents its node location ($x_{i},y_{i}$), Dir represents its node direction ($\theta_{i}$), respectively. Then, go to **step 7**.**Step 5: Decision of Gateway**When $n_{i}$ is between more than one cluster heads, $n_{i}$ will receive more than one **CHI** packet from cluster heads neighbor. Next, $n_{i}$ becomes the gateway node. Otherwise, $n_{i}$ becomes member node, and go to **step 6**.**Step 6: Decision of Cluster Members**Node $n_{i}$ decides the cluster head among the candidates of the cluster head using link stability based on Cosine Similarity and Cosine Distance, as follows. The cosine similarity and cosine distance are used to calculate the link stability between $n_{i}$ and neighbor nodes. The selected cluster member (${\mathrm{CM}}_{m^{*}}$) can be mathematically formulated as:
(7a)$$\begin{array}{cc} m^{⋆} & =\;\arg\max_{m}\left\{\mathrm{CoSim}\left(i,m\right)\right\},m\in {\mathrm{CH}}_{{\mathrm{NB}}_{i}}, \end{array}$$
(7b)$$\begin{array}{cc} s.t. & \;\sum_{m=1}^{M_{\mathrm{NB}}}\mathrm{CoDis}\left(i,m\right)<\hat{\rho }, \end{array}$$
where ${\mathrm{CH}}_{{\mathrm{NB}}_{i}}$ represents a set of cluster heads near $n_{i}$, $M_{\mathrm{NB}}$ represents the number of CMs near node *i*, i.e., $M_{\mathrm{NB}}=\left|{\mathrm{CM}}_{{NB}_{m}}\right|$. ([Disp-formula FD7a-sensors-22-05811]) indicates the maximum cosine similarity between $n_{i}$ and ${\mathrm{CH}}_{m}$, while ([Disp-formula FD7b-sensors-22-05811]) means the cosine distance constraint that must be less than the cosine distance threshold ($\hat{\rho }$). In ([Disp-formula FD7b-sensors-22-05811]), CoSim can be expressed as [39].
(8)$$\begin{array}{c} \mathrm{CoSim}\left(i,m\right)=\frac{\sum_{m=1}^{N}{\vec{V}}_{i}{\vec{V}}_{m}}{\sqrt{\sum_{i=1}^{N}\vec{V_{i}^{2}}}\sqrt{\sum_{m=1,m\neq i}^{N}\vec{V_{m}^{2}}}}, \end{array}$$
where ${\vec{V}}_{i}$ and ${\vec{V}}_{m}$ are the *i*-th and *m*-th node’s vector information, respectively. Each node ${\vec{V}}_{i}$ is related with a mobility vector information metric value (i.e. speed, direction, and location) ${\vec{V}}_{i}=({\vec{V}}_{1}$, ${\vec{V}}_{2}$,⋯,${\vec{V}}_{m})$, where ${\vec{V}}_{i}$ constitutes the vector values which indicate link information between nodes. Cosine similarity can determine the similarity information from each adjacent node, while cosine distance is a method for determining the communication distance between adjacent node. By considering the maximum cosine similarity under the constrained communication distance, we can control the cluster member to make more stable cluster members in the viewpoint of mobility. Then, the cosine distance of the node used to find the distance between two nodes can be calculated by [40]
(9)$$\begin{array}{c} \mathrm{CoDis}(i,m)=\{1-\mathrm{CoSim}(i,m)\}. \end{array}$$If $n_{i}$ selects the ${\mathrm{CH}}_{m^{*}}$, the node ${\mathrm{CH}}_{m^{*}}$ can be as the best cluster head. Node $n_{i}$ sends the joint-cluster **(JC)** packet to the ${\mathrm{CH}}_{m^{*}}$. **JC** packet contains the following fields:
$$\begin{array}{c} \langle \mathrm{Type},S_{\mathrm{ID}},D_{\mathrm{ID}},S,\mathrm{Loc},\mathrm{Dir},\mathrm{Status}\rangle \end{array}$$
where Type represents packet type, $S_{\mathrm{ID}}$ represents source node ID, $D_{\mathrm{ID}}$ represents destination node ID, S represents its node speed ($s_{i}$), Loc represents its node location ($x_{i},y_{i}$), Dir represents its node direction, and Status represents its node status (gateway node or else), respectively. Then, go to **step 7**.**Step 7: Cluster Member Table Updates**Node $n_{i}$ replies the accept **(AC)** packet to the transmitted node and updates the cluster member (CM) table and the cluster has been formed. **AC** packet contains the following fields:
$$\begin{array}{c} \langle \mathrm{Type},S_{\mathrm{ID}},D_{\mathrm{ID}},\mathrm{Status}\rangle \end{array}$$
where Type represents packet type, $S_{\mathrm{ID}}$ represents source node ID, $D_{\mathrm{ID}}$ represents destination node ID, and Status represents its node status (gateway node or else), respectively.−Otherwise, $n_{i}$ waits until it receives **AC** packet.

It is noted that, according to the characteristic of vehicular networks, the nodes dynamically and randomly move. Thus, the members often switch from one cluster to another cluster. However, the member nodes (including the source node and gateway node) do not switch much through the proposed clustering algorithm. The possible reason is that the cluster has a similar mobility pattern via cosine similarity and cosine distance. Besides, since the cluster head is the slowest node, the cluster head does not switch much. Table 1 is summarized the list of packets for the proposed clustering protocol process.

### 4.3. The Proposed Hybrid-Price Auction-Based Secure Routing Protocol: HPA-SR

In this subsection, we explain in detail the proposed hybrid-price auction-based secure routing protocol, named HPA-SR protocol. As can be observed in Figure 4, the source node S begins to establish a routing route to the destination node D.

The event of route establishment happens (on demand-reactive). We propose a HPA-SR protocol that contains the first price auction (FPA) and second price auction (SPA) in which each node employs the Markov decision process (MDP) to select which kind of auction method is used to establish the secure route by avoiding the sinkhole attack, as explained in Section 4.4. Figure 5 illustrates the flowchart of the proposed routing procedure, which can be summarized as follows:


**Route Request Process:**

**Step 1: Initialization**
The source node S starts to establish a routing route between the source node S and the destination node D.
**Step 2: Source Node Operation for Route Request: Generates and Sends RREQ Packet**
If the source node S does not have the routing information to the destination node D, the source node S generates a RREQ packet and sends RREQ packet to the cluster head ${\mathrm{CH}}_{k}$ in its cluster. The RREQ packet contains the following fields:
$$\begin{array}{c} \langle \mathrm{Type},S_{\mathrm{ID}},D_{\mathrm{ID}},S_{\mathrm{Seq}},D_{\mathrm{Seq}},{\mathrm{RREQ}}_{\mathrm{ID}},\mathrm{hop}\rangle \end{array}$$
where Type represents packet type, $S_{\mathrm{Seq}}$ is the source sequence, and $D_{\mathrm{Seq}}$ is the destination sequence, which is the number of attempts to confirm control messages, ${\mathrm{RREQ}}_{\mathrm{ID}}$ is the number of generating RREQ packet on the same session at the source, and hop is denoted as the number of hop to the destination, respectively.
**Step 3: Intermediate Node Operation at Cluster Head for Route Request**
When ${\mathrm{CH}}_{k}$ receives the RREQ packet, ${\mathrm{CH}}_{k}$ records sender’s ID and updates the routing table, then ${\mathrm{CH}}_{k}$ broadcasts RREQ to the gateway node (${\mathrm{GW}}_{k}$) in its cluster or the next cluster heads and goes to **step 4**.
**Step 4: Intermediate Node Operation at Gateway for Route Request**
When the gateway node ${\mathrm{GW}}_{k}$ receives RREQ packet from ${\mathrm{CH}}_{k}$, ${\mathrm{GW}}_{k}$ records sender’s ID and updates the routing table, then ${\mathrm{GW}}_{k}$ broadcasts RREQ to their neighbors node ${\mathrm{NB}}_{k}$ and goes to **step 5**. Otherwise, RREQ packet will be dropped.



**Route Reply Process:**

**Step 5: Destination Node Operation for Route Reply: Generates and Sends RREP Packet**
When ${\mathrm{NB}}_{k}$ is the destination node D, the destination node D records sender’s ID and updates the routing table, then generates a RREP packet. The destination node D unicasts the RREP packet to the previous node. The RREP packet contains the following fields:
$$\begin{array}{c} \langle \mathrm{Type},S_{\mathrm{ID}},D_{\mathrm{ID}},\mathrm{Energy},D_{\mathrm{Seq}},\mathrm{hop}\rangle \end{array}$$
where Energy represents remaining energy of the node. Then, go to **step 6**
**Step 6: Intermediate Node Operation at Previous Node (to the source node) for Route Reply**
When the intermediate node ${\mathrm{NB}}_{j}$ receives RREP packet, ${\mathrm{NB}}_{j}$ records sender’s ID of RREP packet and updates the routing table. Then, go to **step 7**. Otherwise, ${\mathrm{NB}}_{j}$ waits until it receives RREP packet.
**Step 7: Calculation of Cost/Bidding Value for Secure Route Establishment**
The intermediate node ${\mathrm{NB}}_{j}$ calculates cost/bidding value $b_{j}$. ${\mathrm{NB}}_{j}$ will compare the cost/bidding value $b_{j}$ receiving with the bidding threshold $B_{th}$. If the $b_{i}$ is greater than $B_{th}$, ${\mathrm{NB}}_{j}$ will use second price auction (SPA) to determine the route to be pursued by the next node. Otherwise, if the $b_{i}$ is less than $B_{th}$, then ${\mathrm{NB}}_{j}$ will use first price auction (FPA) to determine the route to be traversed by the next node. ${\mathrm{NB}}_{j}$ will select the next node for data transmission based on a hybrid price auction process model that adaptively decides the auction model among the first price auction and the second price auction against the sinkhole attack, then we will obtain the routing table that can be summarized in Table 2, where PN is previous node, ${\mathrm{NB}}_{j}$ as next node NN, Cost is cost/bidding value, $S_{\mathrm{ID}}$ is source ID and $D_{\mathrm{ID}}$ is destination ID, respectively. The routing table will be used to determine the next node to the destination node that will be passed by the data packet during the data transmission process. Besides, we will explain the detailed process of hybrid price auction in Section 4.4. Then, go to data transmission process in **step 8**.



**Data Transmission Process:**

**Step 8: Data Transmission at Source Node**
The source S sends data packet to the destination D based on the routing table, which is determined in **Step 1** to **Step 7**.


**Table 2 sensors-22-05811-t002:** Routing table of the proposed HPA-SR protocol.

PN	NN	Cost	$S_{\mathrm{ID}}$	$D_{\mathrm{ID}}$

The list of packet for the HPA-SR protocol can be summarized in Table 3.

**Figure 5 sensors-22-05811-f005:**
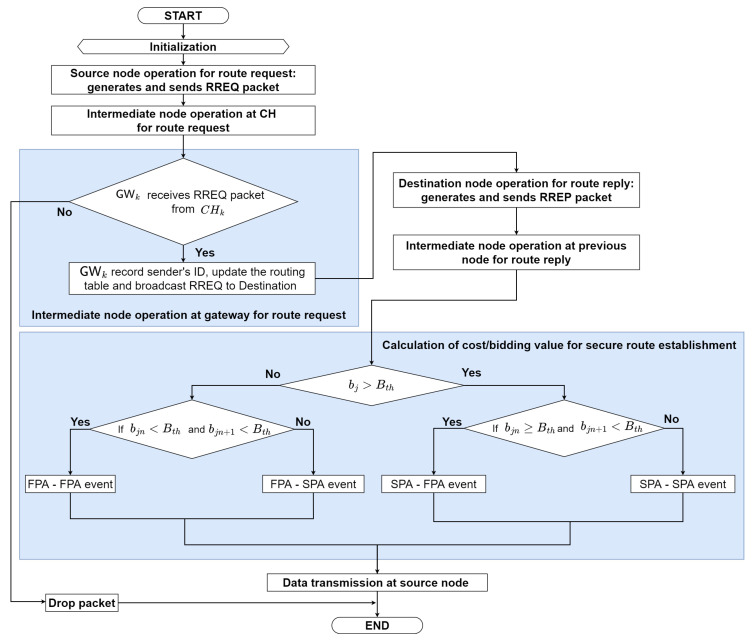
The flowchart of the proposed routing protocol: HPA-SR.

### 4.4. The Hybrid-Price Auction Model Process for The Proposed HPA-SR Protocol

As we can see in Figure 4, a sinkhole attack occurs when a node tries to attack the data of a trusted neighbor by broadcasting fake routing information and then proceeding to the destination node. A sinkhole attack has behavior in choosing the best route in the network. We can solve this problem by using a hybrid-price auction in secure routing to avoid sinkhole attacks. In other words, the source will choose the best route using first or second-price auction to send data transmission to the destination. In this paper, we consider the secure routing protocol based on hybrid-price auction. Therefore, we present the hybrid-price auction model process for the proposed the routing protocol, HPA-SR. In this analysis, we will drive to select the kind of auction model. In secure routing protocol, the sets of players are included as the intermediate nodes and denoted as N=1,2,⋯,n. Thus, the valuation of each nodes is $v_{1}>v_{2}>\cdots >v_{n}>0$. The node valuations of the object are assumed to be all different and all positive. We assume that each intermediate node receives a per-packet cost for forwarding packets and that this cost is private to itself. Each node *i* bids $b_{i}$ its outbound link cost, which is reported during the route discovery process, and each node *i* can only bid one time. We use (Equation 3) as the equilibrium of hybrid-price auction. In summary, the hybrid-price auction is the following strategic game:**Players:** The set I=1,2,⋯,i of the *n* bidders. In this case, players are the intermediate nodes.**Action:** Action is what the player will do. In this case, each player will make a bid.**Payoff:** Since $b_{i}<B_{th}$, the players play the first-price auction. Otherwise, players play the second-price auction. If $b_{i}\geq B_{th}$, we denote $\bar{b}$ as the highest submitted by a player other *i* as in (Equation 4).−If $b_{i}\leq \bar{b}$, the number of each other player who bids $\bar{b}$ is greater than $b_{i}$, then the maximum payoff of the *i*-th player is $v_{i}-\bar{b}$ where $\max_{p_{i}}$ is not equal to zero.

We aim to design an hybrid-price auction that consists of the first-price auction (FPA) and second-price auction (SPA) model to help for determining the route to be chosen by the next node. Based on [35,37] for either FPA or SPA strategies, let $b_{i}$ denote the bidding value of the node, $S_{n}$ represents the session of the network. In this work, the basic concept of the hybrid-price auction can be modeled as a two-state Markov chain with the FPA and SPA states, respectively, as illustrated in Figure 4. In state FPA, the node will use FPA to determine the route to be pursued by the next node, and in state SPA, the node will use SPA to determine the route to be traversed by the next node. The transition of the FPA/SPA strategies can be explained as follows. For illustration, suppose that each node receives an RREP packet from the previous node. We consider remaining energy to get the cost/bidding value because sinkhole nodes and edge servers have unlimited energy while intermediate nodes have limited energy. Thus, we combine the number of hops and the remaining energy of each node to obtain the cost/bidding value. The cost/bidding value is calculated by multiplying the number of hops by the remaining energy of each node, which is expressed as:(10)$$\begin{array}{c} b_{i}={\mathrm{hop}}_{i}\times ER_{i}, \end{array}$$
where ${\mathrm{hop}}_{i}$ represents the number of hops of node *i* and $ER_{i}$ represents remaining energy of node *i* as explained in Section 4.5. Then, the node will compare the cost/bidding value that is received with the cost/bidding threshold. There are two scenarios to compare the cost/bidding value with the cost/bidding threshold. If $b_{i}$ is less than $B_{th}$, the node will use FPA to determine the route to be pursued by the next node. Otherwise, if $b_{i}$ is greater than $B_{th}$, the node will use SPA to determine the route to be traversed by the next node. Accordingly, there are four transition events between the two states as follows:Event 1: The FPA-FPA event: $S_{n}+S_{n+1}$, $b_{in}<B_{th}$ and $b_{in+1}<B_{th}$,Event 2: The FPA-SPA event: $S_{n}+S_{n+1}$, $b_{in}<B_{th}$ and $b_{in+1}\geq B_{th}$,Event 3: The SPA-FPA event: $S_{n}+S_{n+1}$, $b_{in}\geq B_{th}$ and $b_{in+1}<B_{th}$,Event 4: The SPA-SPA event: $S_{n}+S_{n+1}$, $b_{in}\geq B_{th}$ and $b_{in+1}\geq B_{th}$,

Where $S_{n}$ represents the *n*-th session, $S_{n+1}$ represents the next session, $b_{in}$ represents the cost/bidding on node *i* in session *n*, and $b_{in+1}$ represents bidding on node *i* in the next session, respectively. From the transition events, it is noteworthy that: when $b_{in}<B_{th}$, the node will choose the route with the first auction price and when $b_{in}\geq B_{th}$, the node will choose the route with the second auction price, regardless of the conditions on each session. For steady-state probabilities, let $p_{00}$, $p_{01}$, $p_{10}$, and $p_{11}$ denote the transition probabilities of events one to four, respectively. Let $\pi_{1}$ and $\pi_{0}$ denote the steady-state probabilities of the FPA and SPA status, respectively. The relationship between $\pi_{1}$ and $\pi_{0}$ associated with the described Markov chain can be expressed as [41]
(11)$$\begin{array}{cc} \pi_{0} & =p_{00}\pi_{0}+p_{10}\pi_{1},\\ \pi_{1} & =p_{01}\pi_{0}+p_{11}\pi_{1},\\ 1 & =\pi_{0}+\pi_{1}. \end{array}$$

Relying on the fact that $p_{00}=1-p_{01}$ and $p_{11}=1-p_{10}$, and after some modifications, $\pi_{0}$ and $\pi_{1}$ can be written as
(12)$$\begin{array}{cc} \pi_{0} & =\frac{p_{10}}{1-p_{00}+p_{10}}=\frac{p_{10}}{p_{01}+p_{10}}, \end{array}$$
(13)$$\begin{array}{cc} \pi_{1} & =\frac{p_{01}}{1-p_{11}+p_{01}}=\frac{p_{01}}{p_{01}+p_{10}}. \end{array}$$

### 4.5. Energy Consumption Model

In this paper, we used an energy model to calculate the energy consumption for sending and receiving packet over a link [42,43]. As we can see in Figure 6, the energy model will be used for the HPA-SR to calculate the residual energy required by the sender and receiver to send a number of packets. Therefore, a node can choose the next node to send data so that the residual energy of the sender and receiver is greater than the energy threshold, thereby extending the life of the route. We consider that all nodes are equipped with IEEE 802.11a 11 Mbps network interface card, whose electric currents are 280 mA and 330 mA in reception mode and transmission mode, respectively, and the electric potential is 5 V [44]. The remaining energy $ER_{i}$ at node *i*-th can be formulated as:(14)$$\begin{array}{c} ER_{i}=ER_{pi}-E_{\mathrm{mode}}\;\left[\mathrm{joule}\right], \end{array}$$
where $ER_{pi}$ is the current remaining energy of the node *i*-th, and $E_{\mathrm{mode}}$ is energy consumption model that has transmit or receive modes. The energy consumption model when node *i*-th transmits and receives data packet p is expressed mathematically as follows.
(15)$$\begin{array}{cc} E_{\mathrm{Tx}}\left(i,p\right) & =I_{\mathrm{tx}}\times V\times t_{p}\;\left[\mathrm{joule}\right], \end{array}$$
(16)$$\begin{array}{cc} E_{\mathrm{Rx}}\left(i,p\right) & =I_{\mathrm{rx}}\times V\times t_{p}\;\left[\mathrm{joule}\right] \end{array}$$
where $I_{\mathrm{tx}}$ and $I_{\mathrm{rx}}$ represent the electric currents of transmission and reception, respectively. *V* is the electric potential, and $t_{p}$ represents the time taken to transmit the packet p (in seconds).

## 5. Performance Evaluation

### 5.1. Simulation Environments and Parameters

In the simulation, to provide more insight into the proposed routing protocol and clustering, we compared the performance of the HPA-SR protocol with AODV routing protocol. The simulation environments and parameters are presented in Table 4. In particular, we deployed 30, 50, and 100 nodes moving over an area of size 1000 × 1000 m^2^ in urban scenario. In this case, mobile nodes move according to the group mobility. The nodes are divided into groups, and several groups will move in the same direction. The groups will build their movements based on the group leader’s movement [45]. The initial position of nodes is randomly distributed along the street and moves with different speeds (20 km/h, 40 km/h, 60 km/h, and 80 km/h) [46]. The MAC layer is modeled using the IEEE 802.11 standard, and using the received signal strength indicator (RSSI) threshold is −80 dBm for communication range to more practically. One of the reasons for considering the use of RSSI is that the value of fluctuations in RSSI obtained has taken into account its effect on changes in channel conditions including multi-path fading [47]. All the simulation experiments are carried out on the NS3 simulator.

### 5.2. Performance Metrics

The performances of the proposed routing and the clustering protocols, HPA-SR and ASCS, are evaluated in terms of the following metrics:Packet delivery ratio (PDR): it is defined by the ratio of the number of the received data packet at the destination node over the number of the transmitted data packet at the source node.Delay: it is defined by the average latency to establish the route per one session.Control overhead: it is defined by the average number of control packets to establish a route per session per node.The average number of the cluster head change: it is defined by the average number of cluster heads changes in per cluster per session [33].Packet loss ratio: it is defined by the ratio of the number of packets loss to the total number of sent packets [48].

### 5.3. Numerical Results

In this subsection, we present illustrative simulation results for the achievable performance of the proposed HPA-SR protocol approach. In particular, we set in the simulations parameter as shown in Table 4. We use the NS3 simulation program, where the algorithm is run with 200 s with 5 s each sessions. The simulation results in every figure are obtained with an average of 40 independent session.

To illustrate the effectiveness of suggested algorithm (HPA-SR with ASCS), we will compare the performance of the suggested algorithm with AODV protocol (with or without combining ASCS clutering protocol).

Figure 7 represents the comparison of the average number of cluster head changes in each session as a function of node speed to evaluate cluster stability. As can be seen in Figure 7, when the node speed increases, the average number of cluster head change is increased. One of possible reasons is that when the node speed is increased, the node location is changed frequently, which causes the broken of clustering. In addition, when the number of node in the network increases, the average number of cluster head change is increased. It can be explained as when the density of networks increase, the cluster head is changed. Nevertheless, the average number of cluster head change is less than one. It means that the number of cluster head change is less than one in each session. Thus, the proposed clustering is very stable.

Figure 8 shows the comparison of the packet delivery ratio as a function of node speed. As can be seen in Figure 8, when the node speed increases, the packet delivery ratio is decreased. One of the possible reasons is that when the node speed is increased, the entire network becomes more unstable and dynamic as velocity reaches higher values, which causes packet loss. However, we can notice that the decrease in packet delivery ratio is significantly less in the case of HPA-SR with ASCS (HPA-SR+ASCS) protocol than in other cases. Thus, HPA-SR with ASCS protocol proved to be the most reliable in terms of packet delivery ratio.

Figure 9 presents the comparison of the routing delay including latency time for cluster construction per session as a function of node speed. As can be seen in Figure 9, the pattern of routing delay is shown to increase when the node speed increases. This result can be explained as the route establishment spending more time because the node moves more dynamically. However, the proposed HPA-SR with ASCS (HPA-SR+ASCS) protocol only involves the CH and GW nodes to determine the route to be traversed by the packet. Thus, the HPA-SR+ASCS protocol can send packets with a minimum delay compared to other protocols.

Figure 10 represents the comparison of the control overhead including the control overhead for cluster construction per node per session as a function of node speed. As shown in Figure 10, when the speed increases, the control overhead is increased little bits, but not significant. One of the possible reasons is that if node speed increases, the distance is increased, which causes the need for more packets to establish the route. However, ASCS clustering protocol can reduce the control overhead in our proposed routing protocol compared with other cases. It means that ASCS clustering protocol only involves CH and GW in the routing process and the decrease in control overhead is significantly less in case HPA-SR+ASCS protocol than in other cases. Thus, the HPA+ASCS protocol demonstrated that it can improve connectivity while also being the most stable in terms of control overhead.

Now, we turn to our attention to security perspective. Figure 11 shows the comparison of the average packet loss ratio Figure 11a and number of packet loss Figure 11b in each session as a function of node speed in km/h, respectively. The average packet loss ratio shown in Figure 11a is defined by the ratio of the number of loss packets to the total number of sent packets. As can observed in Figure 11a, when the node speed increases, the average packet loss ratio is increased. At the same time in Figure 11b, when the node speed increases, the average number of packet loss is increased. One of the possible reasons is that when the node speed increase, the location is changed frequently, which causes the packet to be sent directly to the sinkhole node. However, we can notice that the clustering can reduce the number of links between nodes. Thus, the proposed HPA-SR+ASCS protocol is proved to be secure in terms of network security perspective.

Finally, we exploit the impact of the number of nodes on the network metrics. In Figure 12, we evaluate the packet delivery ratio in the effect of the number of node on the proposed routing protocol with the proposed clustering as a function of node speed with a different number of nodes. As we can see in Figure 12, when the node speed is increased, the PDR will decrease a little bit. Besides, when the number of node increases, the PDR will be increased little bit. The reason is that when the number of node increases, the density of network is increased, which provides more strong connectivity between nodes. The HPA-SR+ASCS protocol with number of nodes N = 100 outperforms with the high packet delivery ratio.

Figure 13 presents the routing delay in the effect of the number of node on the proposed routing protocol with the proposed clustering as a function of node speed with different number of nodes. As can be seen in Figure 13, when the node speed and the number of node increases, the routing delay is increased little bits, but not significantly. It can be explained that when the number of node and speed increased, the number of hop is increased, which causes the routing process to take longer.

Figure 14 presents the control overhead as function of node speed with different number of nodes on HPA-SR+ASCS protocol with different number of nodes. As can be observed in Figure 14, when the speed and number of nodes increases, the control overhead is increased. There are two possible reasons as follows. Firstly, when the number of node increases, the density of the network is increased, which leads to the increasing of RREQ and RREP packet transmission frequency. Secondly, when the number of node speed increases, since the node is more easily broken, the control overhead is increased.

## 6. Conclusions

In this paper, we proposed a hybrid-price auction-based secure routing (HPA-SR) protocol and an advanced speed and cosine similarity-based clustering (ASCS) protocol as an underlying structure to establish a secure route against sinkhole attacks and improve connectivity between nodes. The proposed HPA-SR protocol used the first- and second-price auction to avoid sinkhole attacks. Each node was used in the Markov decision process to conditionally select which kind of auction method establishes the secure route against the sinkhole attack. Besides, to improve connectivity between nodes, the proposed ASCS clustering protocol that works as underlying structure used the node’s speed and direction and then calculated the cosine similarity and distance between nodes. The numerical results showed that the use of hybrid-price auction and advanced speed cosine similarity improves the performance of routing in the network. The proposed HPA-SR with ASCS outperforms either the AODV+ASCS, HPA-SR, or AODV protocol in terms of the security in the network and packet delivery ratio. Additionally, the proposed HPA-SR with ACSS protocol are able to reduce the routing delay, packet loss ratio, number of packet loss, and control overhead. 

## Figures and Tables

**Figure 1 sensors-22-05811-f001:**
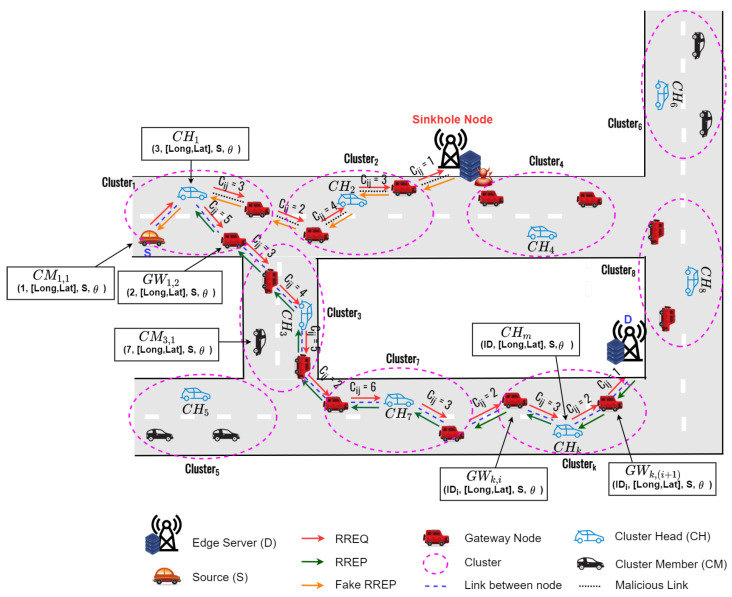
The basic concepts of the proposed routing protocol: HPA-SR.

**Figure 2 sensors-22-05811-f002:**
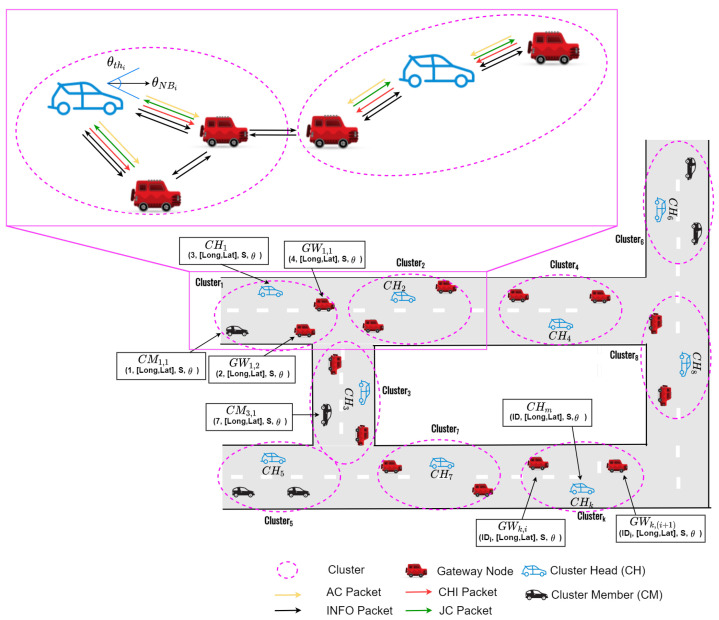
The basic concepts of the ASCS clustering protocol.

**Figure 3 sensors-22-05811-f003:**
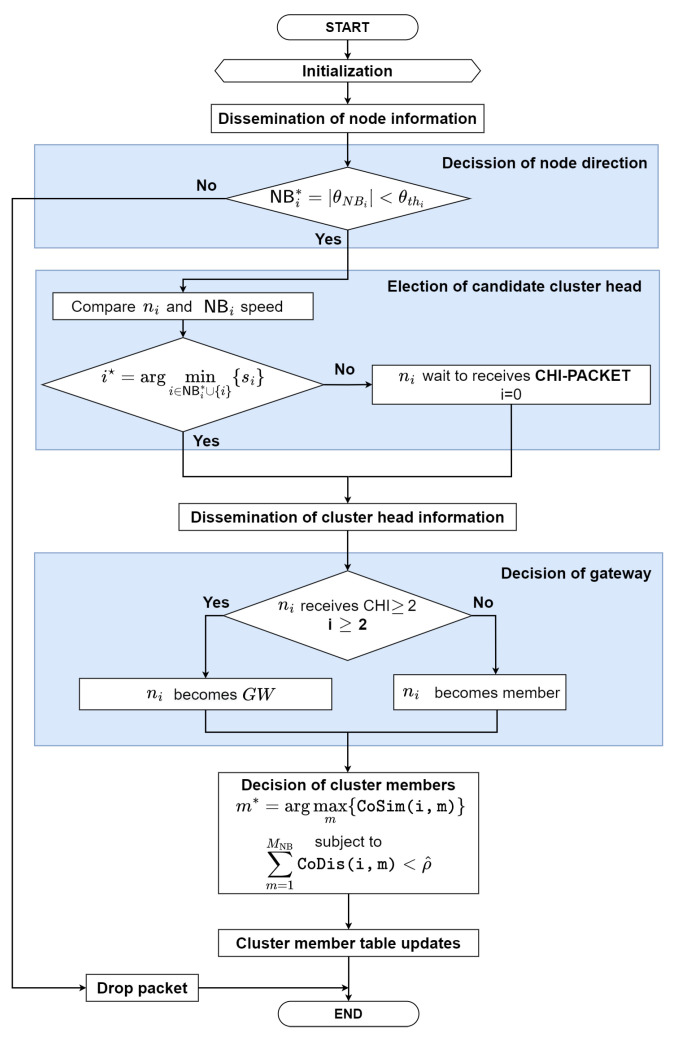
The flowchart of the proposed clustering protocol: ASCS.

**Figure 4 sensors-22-05811-f004:**
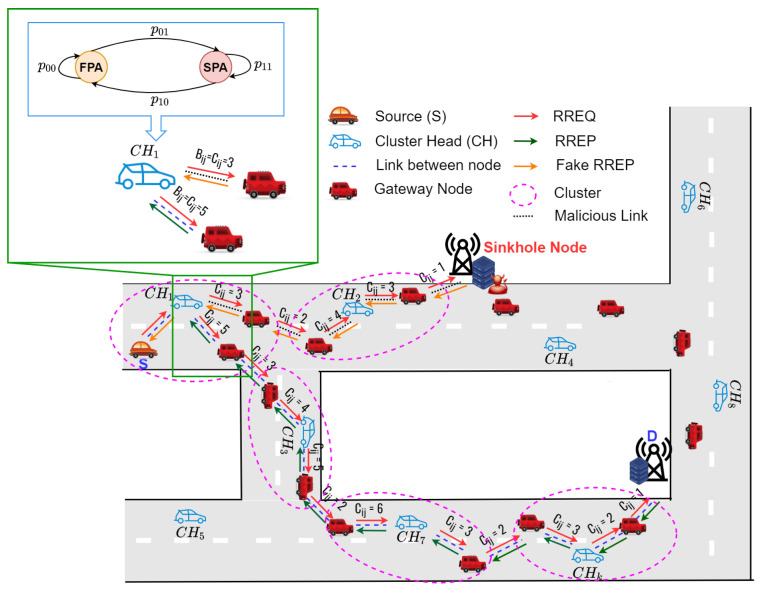
The proposed routing protocol: HPA-SR.

**Figure 6 sensors-22-05811-f006:**
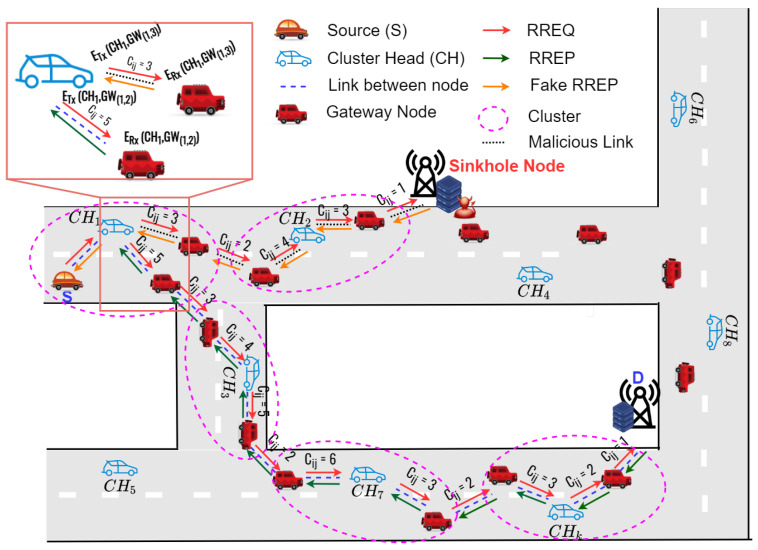
The basic concepts of the energy consumption model for the HPA-SR.

**Figure 7 sensors-22-05811-f007:**
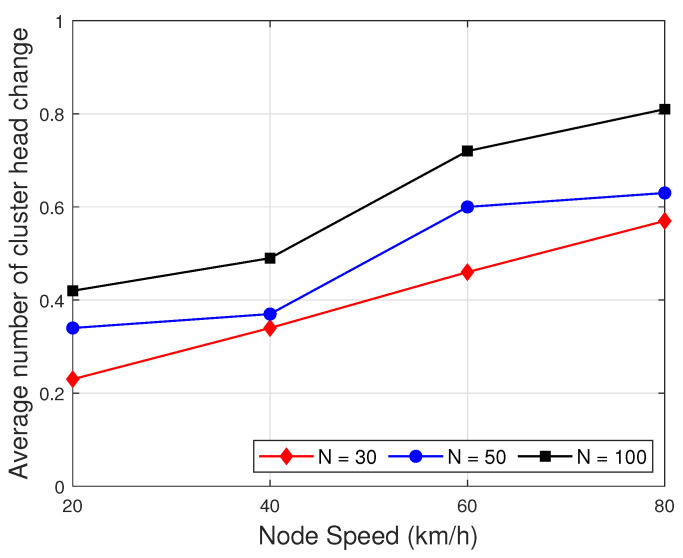
Average number of the cluster head change as a function of node speed.

**Figure 8 sensors-22-05811-f008:**
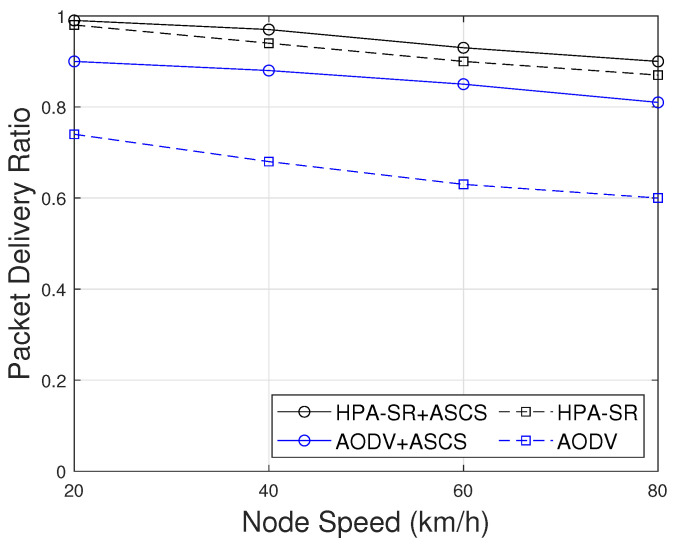
PDR as a function of node speed with difference scenarios.

**Figure 9 sensors-22-05811-f009:**
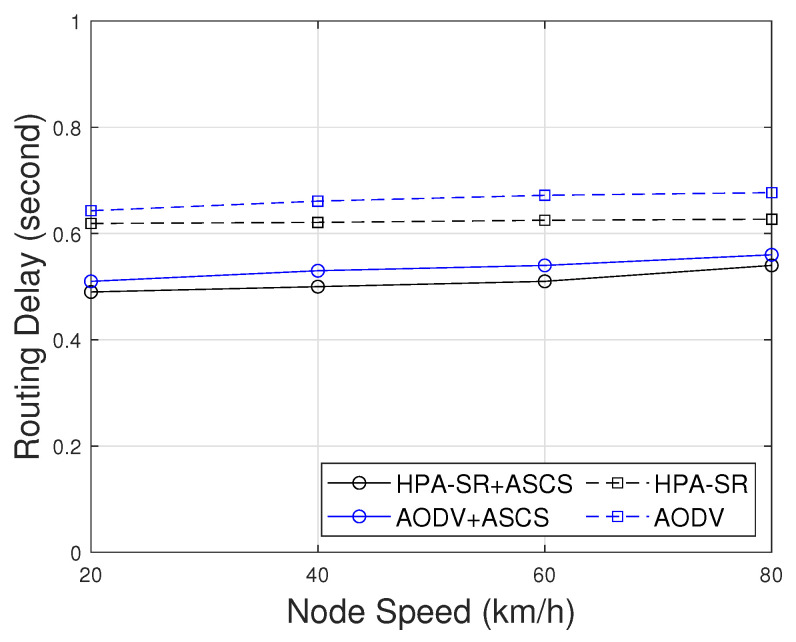
Routing delay as a function of node speed with different scenarios.

**Figure 10 sensors-22-05811-f010:**
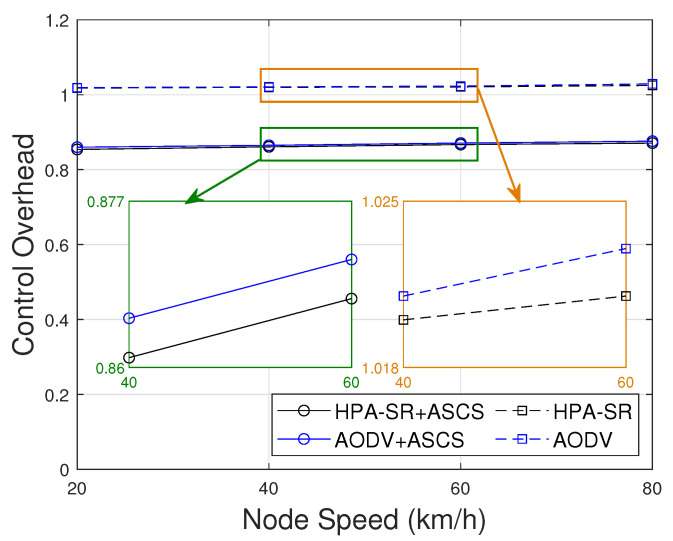
Control overhead as function of node speed with difference scenarios.

**Figure 11 sensors-22-05811-f011:**
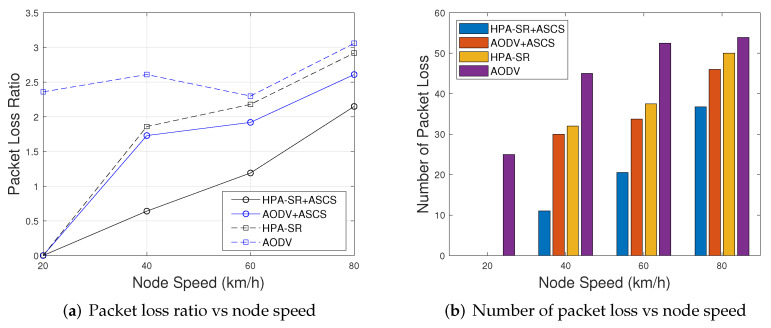
The comparison of packet loss ratio and number of packet loss as a function of node speed with different scenarios.

**Figure 12 sensors-22-05811-f012:**
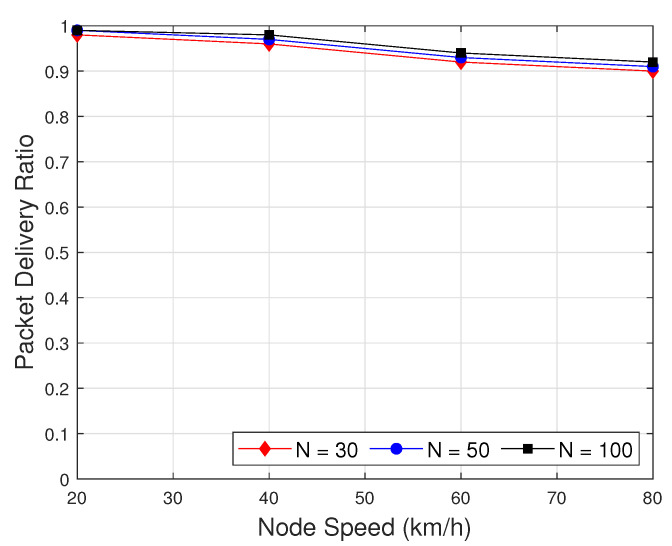
Packet delivery ratio as function of node speed with different number of nodes.

**Figure 13 sensors-22-05811-f013:**
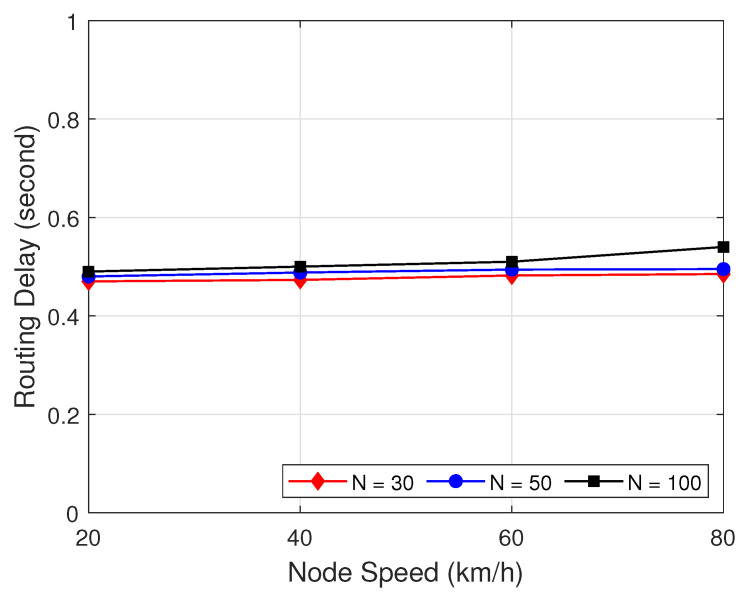
Routing delay as a function of node speed with different number of nodes.

**Figure 14 sensors-22-05811-f014:**
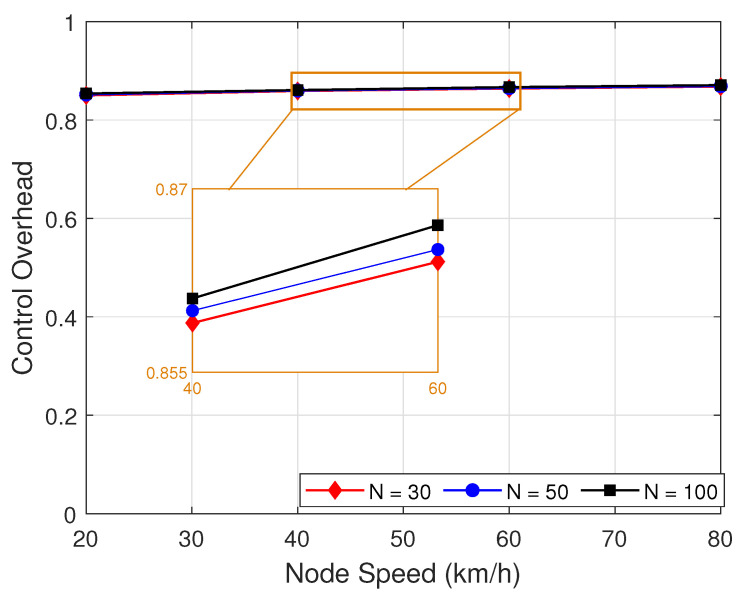
Control overhead as function of node speed with different number of nodes.

**Table 1 sensors-22-05811-t001:** List of Packets for The ASCS clustering protocol.

Packet Name	Full Name	Field Information
INFO	Information Packet	$\mathrm{Type},S_{\mathrm{ID}},D_{\mathrm{ID}},S,\mathrm{Dir}$
CHI	Cluster Head Info Packet	$\mathrm{Type},S_{\mathrm{ID}},D_{\mathrm{ID}},S,\mathrm{Loc},\mathrm{Dir}$
JC	Joint-Cluster Packet	$\mathrm{Type},S_{\mathrm{ID}},D_{\mathrm{ID}},S,\mathrm{Loc},\mathrm{Dir},\mathrm{Status}$
AC	Accept-Cluster Packet	$\mathrm{Type},S_{\mathrm{ID}},D_{\mathrm{ID}},\mathrm{Status}$

**Table 3 sensors-22-05811-t003:** List of packets for the HPA-SR protocol.

Packet Name	Full Name	Field Information
RREQ	Route request	$\mathrm{Type},S_{\mathrm{ID}},D_{\mathrm{ID}},S_{\mathrm{Seq}},D_{\mathrm{Seq}},{\mathrm{RREQ}}_{\mathrm{ID}},\mathrm{hop}$
RREP	Route reply	$\mathrm{Type},S_{\mathrm{ID}},D_{\mathrm{ID}},\mathrm{Energy},D_{\mathrm{Seq}},\mathrm{hop}$

**Table 4 sensors-22-05811-t004:** Simulation Environments and Parameters.

Parameters	Value
Simulator	NS3
Simulation area	1000 × 1000 $m^{2}$
Packet size	1024 bits
Mobility model	Group Mobility
Radio range	250 m
Simulation time	200 s
Session length	5 s
Number of nodes	[30, 50, 100]
Node’s Speed Range	[20:20:80] (km/h)
Receive signal strength indicator (RSSI) threshold	−80 dBm
MAC protocol	802.11a

## Data Availability

Not applicable.

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
