# Peer review of "A Hybrid Price Auction-Based Secure Routing Protocol Using Advanced Speed and Cosine Similarity-Based Clustering against Sinkhole Attack in VANETsâ€"

_sensors, 2022, doi:10.3390/s22155811_

Round 1
Reviewer 1 Report
As per the section 4.1, The proposed clustering (ASCS) protocol considers the speed and direction of node, how the proposed algorithm will handle the situation in case of extreme outer edge of the cluster and also in case of overlap between two clusters ?
Rest of the analysis looks fine to be. You may explore few transactions/ high index journals and conferences from year 2022. Few of the recommended references are as;
(a)Sensors | Free Full-Text | An Intelligent Opportunistic Routing Algorithm for Wireless Sensor Networks and Its Application Towards e-Healthcare (mdpi.com)
(b)An intelligent trust sensing scheme with metaheuristic based secure routing protocol for Internet of Things | SpringerLink
(c)A blockchain-based secure routing protocol for opportunistic networks | SpringerLink
(d)A Trust Based Secure Intelligent Opportunistic Routing Protocol for Wireless Sensor Networks | SpringerLink
Few more papers should also be look into. Accordingly the related work should be modified.
Author Response
Dear Reviewer,
We would like to thank the anonymous Reviewers for their constructive comments which greatly help us to improve the quality of this manuscript.
In the uploaded file, we provide a point-by-point response and explanation to Reviewers’ comments in detail, with the Reviewers’ comments appearing in italic font (with blue color) and our answers in normal font (with black color). Unless stated otherwise, all indexed items (e.g., figures, equations, etc.) in the response letter refer to those in the revised manuscript.
Furthermore, in the revised manuscript, the major changed parts have been highlighted in normal font (with blue color). Other minor corrections and rearrangements have not been highlighted. Please noted that, in this response, we provide many references to support our explanation and background acknowledge. However, in the revised manuscript, we only cite very closely references.
Finally, we hope that the Reviewer find our explanation/justifications and revisions/improvements satisfactorily.
Sincerely,
The Authors

Reviewer 2 Report
The authors propose a hybrid-price auction-based secure routing protocol using advanced speed and cosine similarity-based clustering to establish a secure route to avoid sinkhole attacks and improve connectivity between nodes. In my opinion, this work is meaningful for network and secure. There is one comment below which I recommend to give one chance to take a revision. A more comprehensive literature survey may be provided and compared with BLCS: brain-like distributed control security in cyber physical systems, distributed blockchain-based trusted multidomain collaboration for mobile edge computing in 5G and beyond, blockchain-enabled tripartite anonymous identification trusted service provisioning in industrial IoT. What is the difference between the proposal and other schemes.
Author Response

(The authors gave the same response as above.)

Reviewer 3 Report
The paper proposes a hybrid auction based routing algorithm that can be used in vehicular networks. Speed and cosine based similarity metrics are used to develop the clustering algorithm. A Markov Decision Process based auction mechanism is incorporated in the protocol.
Comments:
-The advantage of using auction based scheme can be highlighted further.
-Moreover, is there any tradeoff of using auction based schemes as compared to other techniques in literature.
-It can be clarified that how the proposed protocol works in dynamic channel conditions and high mobility scenarios, which are key features of vehicular networks.
- Literature review can be further improved by adding about clustering techniques related to VANETs such as "A Multi-hop Broadcast Protocol Design for Emergency Warning Notification in VANETs".
-How does the behavior of protocol changes in case of multi-path fading? As multi-path fading is important characteristic of vehicular network, what changes in the protocol are required. A small discussion on it can be added.
Author Response

(The authors gave the same response as above.)

Reviewer 4 Report
The authors propose a method to avoid sinkhole attacks based on the HPA-SR protocol, and use APCS Clustering method to enhance the connectivity between nodes. The HPA-SR with ASCS protocol has better performance than the benchmarking.
The authors need to describe more about the impact of sinkhole and present in detail the process by which secure routes are established when members of one cluster switch to other clusters.
.
Author Response

(The authors gave the same response as above.)

Round 2
Reviewer 3 Report
Authors have addressed comments of the last round and I suggest acceptance of the paper.
Author Response
Dear Reviewer 3, We would like to thank the anonymous Reviewer 3 for his/her constructive comments which greatly help us to improve the quality of this manuscript. In the following, we provide a point-by-point response and explanation to Reviewer 3' comments in detail, with the Reviewer 3' comments appearing in italic font (with blue color) and our answers in normal font (with black color). Unless stated otherwise, all indexed items (e.g., figures, equations, etc.) in the response letter refer to those in the revised manuscript. Furthermore, in the revised manuscript, the major changed parts have been highlighted in normal font (with blue color). Other minor corrections and rearrangements have not been highlighted. Please noted that, in this response, we provide many references to support our explanation and background acknowledge. However, in the revised manuscript, we only cite very closely references. Finally, we hope that the Reviewer finds our explanation/justifications and revisions/improvements satisfactorily. Yours sincerely,The Authors

Reviewer 4 Report
The authors need to describe more about the impact of sinkhole and present in detail the process by which secure routes are established when members of one cluster switch to other clusters.
Author Response
Dear Reviewer 4,
We would like to thank the anonymous Reviewer 4 for his/her constructive comments which greatly help us to improve the quality of this manuscript.
In the following, we provide a point-by-point response and explanation to Reviewer 4’ comments in detail, with the Reviewer 4’ comments appearing in italic font (with blue color) and our answers in normal font (with black color). Unless stated otherwise, all indexed items (e.g., figures, equations, etc.) in the response letter refer to those in the revised manuscript.
Furthermore, in the revised manuscript, the major changed parts have been highlighted in normal font (with blue color). Other minor corrections and rearrangements have not been highlighted. Please noted that, in this response, we provide many references to support our explanation and background acknowledge. However, in the revised manuscript, we only cite very closely references.
Finally, we hope that the Reviewer finds our explanation/justifications and revisions/improvements satisfactorily.
Yours sincerely,
The Authors
